# Single-molecule detection with a millimetre-sized transistor

Eleonora Macchia[1], Kyriaki Manoli [1], Brigitte Holzer [1], Cinzia Di Franco[2], Matteo Ghittorelli [3], Fabrizio Torricelli [3], Domenico Alberga[4], Giuseppe Felice Mangiatordi[4,8], Gerardo Palazzo [1,5], Gaetano Scamarcio [2,6] & Luisa Torsi [1,5,7]

Label-free single-molecule detection has been achieved so far by funnelling a large number of ligands into a sequence of single-binding events with few recognition elements host on nanometric transducers. Such approaches are inherently unable to sense a cue in a bulk milieu. Conceptualizing cells' ability to sense at the physical limit by means of highly-packed recognition elements, a millimetric sized field-effect-transistor is used to detect a single molecule. To this end, the gate is bio-functionalized with a self-assembled-monolayer of $10^{12}$ capturing anti-Immunoglobulin-G and is endowed with a hydrogen-bonding network enabling cooperative interactions. The selective and label-free single molecule IgG detection is strikingly demonstrated in diluted saliva while 15 IgGs are assayed in whole serum. The suggested sensing mechanism, triggered by the affinity binding event, involves a work-function change that is assumed to propagate in the gating-field through the electrostatic hydrogen-bonding network. The proposed immunoassay platform is general and can revolutionize the current approach to protein detection.

[1] Dipartimento di Chimica, Università degli Studi di Bari "Aldo Moro", 70125 Bari, Italy. [2] CNR, Istituto di Fotonica e Nanotecnologie, Sede di Bari, 70125 Bari, Italy. [3] Dipartimento Ingegneria dell'Informazione, Università degli Studi di Brescia, 25123 Brescia, Italy. [4] Dipartimento di Farmacia, Scienze del Farmaco, Università degli Studi di Bari "Aldo Moro", 70125 Bari, Italy. [5] CSGI (Centre for Colloid and Surface Science), 70125 Bari, Italy. [6] Dipartimento di Fisica "M. Merlin", Università degli Studi di Bari – "Aldo Moro", 70125 Bari, Italy. [7] The Faculty of Science and Engineering, Åbo Akademi University, 20500 Turku, Finland. [8] Present address: Istituto Tumori IRCCS Giovanni Paolo II, Viale O. Flacco 65, 70124 Bari, Italy. Correspondence and requests for materials should be addressed to L.T. (email: luisa.torsi@uniba.it)

Many systems in nature interact at the single-molecule level. Sea urchins' sperm cells sense environmental cues down to the physical limit to quickly find the oocyte[1]. Neurons can track single pheromones[2] while rod cells on the retina respond to single photons[3,4]. To assure a high interaction cross-section, a large number of highly-packed receptors are generally engaged[5,6].

Label-free single-molecule detection has up until now been achieved with nanometre-sized transducers. Paradigmatic is the single DNA strand detection by means of few bio-probes attached to a single-nanotube transistor[7]. A plasmon-enhanced-field generated in gold nanorods[8] and nanopores[9,10], as well as nanoscopic force-spectroscopies[11,12], were also proven capable of single-molecule label-free detection. Apparently, the current approaches, relying on a nano-transducer hosting few bio-receptors, are unable to sense a cue in a bulk milieu[13] because the interaction cross section is negligibly small. Furthermore, the fabrication scalability of nano-transducers can be challenging. Printable bioelectronics[14] show promises for healthcare and human well-being[15–18]. Electrolyte-gated organic-field-effect-transistors (EGOFETs),[19,20] in particular, are sensors[21,22] endowed with selectivity by the integration of bio-recognition elements[21,23,24]. Their sensitivity has so far enabled a detection limit of 40 aM (10$^{-18}$ moles l$^{-1}$, M) or equivalently of $2.4 \times 10^3$ molecules in 100 µl[25].

Here we report a label-free, single-molecule detection platform based on an EGOFET immobilizing ~$10^{12}$ anti-human-Immunoglobulin-G (anti-IgG) capturing antibodies on its millimetre-sized gate[26], and demonstrate the selective detection of a single human-IgG in diluted saliva (L. Torsi et al. "A field-effect transistor sensor." European Patent Application no. 17177349.2, filed 2017) and 15 ± 4 IgGs in whole serum. This single-molecule transistor (SiMoT) platform shows in both cases world record detection limits in label-free assays and holds the potential to set the ground for a revolution in protein detection and bio-markers label-free assay for early medical diagnostics.

## Results

**Label-free proteins detection at the physical limit in serum**. Figure 1a, b shows the SiMoT comprising a gold gate modified with a self-assembled monolayer (SAM) of the capturing anti-IgGs and a P3HT organic semiconductor (OSC) which forms the FET-channel while water serves as the gating electrolyte. Typical SiMoT output ($I_D - V_D$) and transfer ($I_D - V_G$) characteristic curves are shown in Supplementary Fig. 7a, b (Supplementary Note 4), respectively.

The SAM, featured in Fig. 1c, comprises a chemical (chem-SAM) and a biological (bio-SAM) SAM. The former is composed of mixed alkanethiols endowed with carboxylic terminal groups (3-MPA and 11-MUA, 10:1) that spontaneously self-assemble on a gold surface. After the EDC/sulfo-NHS chemical activation of the chains' carboxylic groups, the capturing anti-IgGs are conjugated to the chem-SAM, by immersing the gate in an anti-IgG solution[27]. The layer of anti-IgGs attached to the chem-SAM forms the bio-SAM. To confer chemical stability to the SAM, the blocking of unreacted carboxylic groups with ethanolamine was carried out. Relevantly, this also generates a network of hydrogen bonds that tightens the shorter chains into a packed monolayer[28] and is likely capable of sustaining electro-static cooperative interactions.

Both the chemical activation and the mixed lengths of the chains in the chem-SAM[29] are known to promote the assembly of

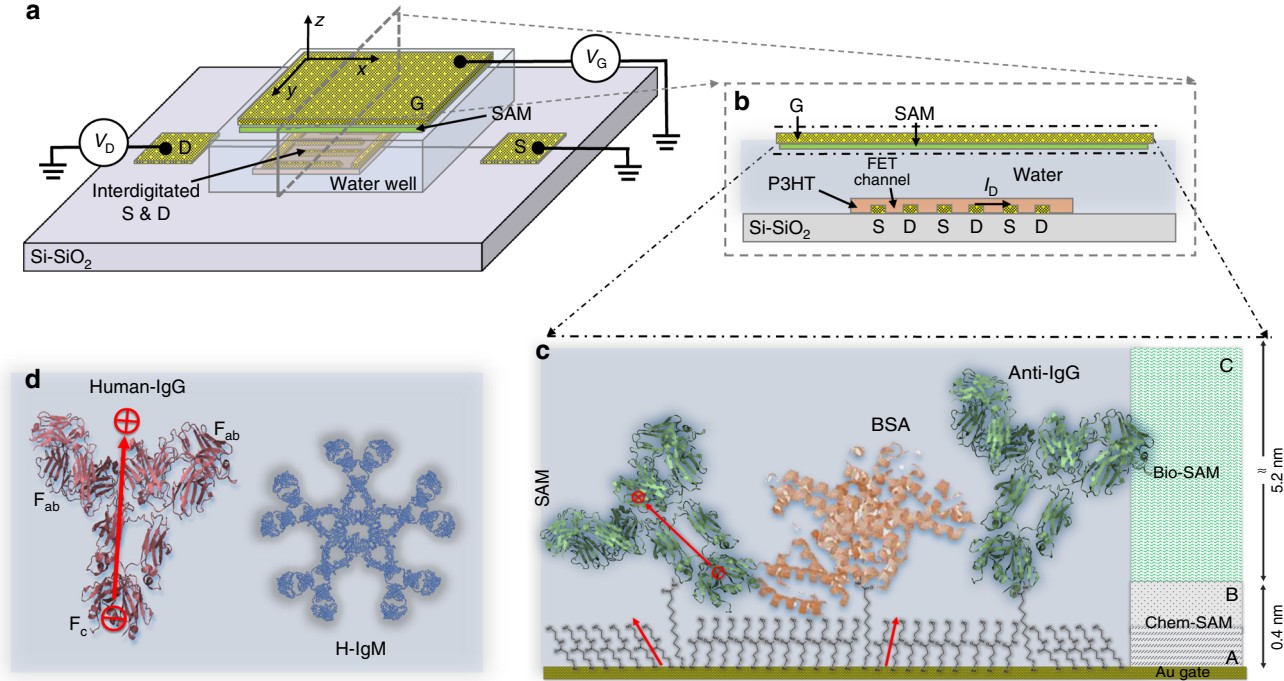

**Fig. 1** The SiMoT electrolyte-gated OFET. **a** The three-dimensional schematic structure of the FET. **b** The cross-sectional structure of the transistor channel region. The source (S) and drain (d) interdigitated contacts are defined on a Si/SiO₂ substrate and covered by a spin coated p-type, Poly(3-hexylthiophene-2,5-diyl)-P3HT, organic semiconductor (OSC) layer. A 300 µl volume of HPLC-grade water is dispensed into a polydimethylsiloxane well covering the OSC surface. A bio-functionalized Au-gate stably hangs over the device at a distance of ~4 mm from the OSC surface while in contact with the water. The capturing SAM, comprising both a chem-SAM of activated-and-blocked 3-mercaptopropionic acid (3-MPA) and 11-mercaptoundecanoic acid (11-MUA) and a bio-SAM of capturing proteins, is sketched in (**c**). For the sake of clarity, features are not in scale. The dipole moments associated with the different components are schematically depicted with red arrows. The whole SAM thickness is 5.6 nm (Supplementary Note 2) and a surface characterization of the SAM is reported in Supplementary Note 3. The structure of IgG and IgM ligands are depicted in (**d**)

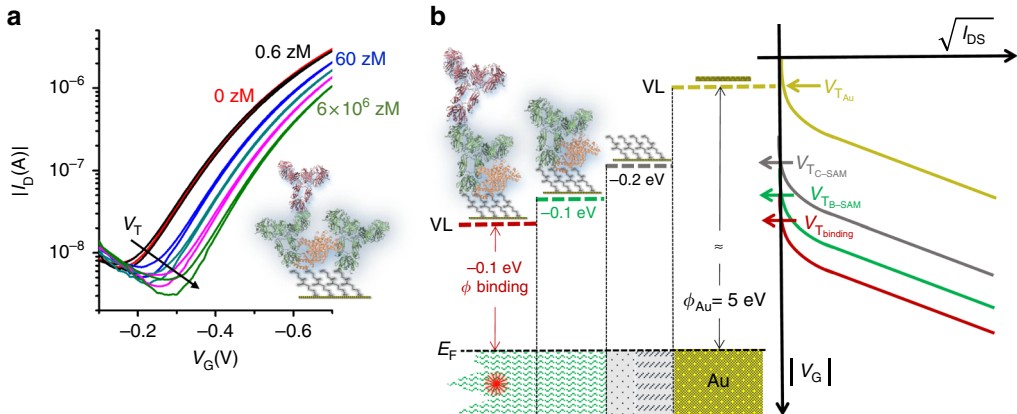

**Fig. 2** Sensing measurements. **a** The SiMoT transfer characteristics ($I_D$ vs. $V_G$ at $V_D = -0.4$ V). The red curve corresponds to the anti-Human-IgG SAM incubated in the bare PBS solution. The same gate is further exposed, in sequence, to PBS standard solutions of IgG at concentrations of 6 zM (black curve), $(6 \pm 3) \times 10$ zM (blue curve), $(6 \pm 1) \times 10^2$ zM (dark curve), $(6.7 \pm 0.1) \times 10^3$ zM (magenta curve) and $(6.67 \pm 0.01) \times 10^6$ zM (light green curve). The decrease of the gate work-function ($\phi$) at the different functionalization steps and after the IgG binding is schematically depicted in (**b**) and correlated to the measured $V_T$ shifts towards more negative gate voltages. $E_F$ is the gate electrochemical potential while VL is the vacuum level. The $V_G$ axis is directed towards negative values and the measured $I_D$ values are negative

a dense layer of proteins[30]. Hence, finally, a bio-SAM composed of $\sim 10^{12}$ capturing antibodies was produced (Supplementary Note 1). The anti-IgGs are packed on the gate at a density of $10^4\ \mu m^{-2}$ which is comparable to that of guanylyl cyclase receptors present on a sperm cell or rhodopsins in photoreceptors[3,6], both capable of single entity detection. The bio-SAM comprises also an aliquot of physisorbed bovine serum albumins (BSAs, Fig.1c) that minimizes non-specific binding and further compacts the anti-IgG bio-SAM by filling the voids left after the anti-IgG conjugation (Supplementary Notes 1, 2). This is confirmed by the evidence that the overall SAM thickness (5.6 nm) does not change after the BSA deposition (Supplementary Note 1).

Human-IgG (Fig.1d) is the affinity ligand that selectively binds to the anti-Human-IgG. Human-IgM (Fig.1d) and BSA, that do not bind to the anti-Human-IgG, serve in negative control experiments. Endogenous bovine-IgG in bovine serum does not bind anti-Human-IgG either.

The IgG detection is performed by measuring the SiMoT transfer characteristics after incubation of the SAM in 100 μl of phosphate buffer saline (PBS) standard solutions of IgG ranging from $6 \times 10^{-2}$ zM to $6 \times 10^8$ zM nominal concentrations. The PBS solution reproduces a physiologically relevant environment having a pH of 7.4 and an ionic-strength of 162 mM, that are both characteristic of blood serum. The standard solutions dilution and sampling errors are given in Supplementary Table 3 and described in Supplementary Note 5.

Typical sensing transfer characteristics, measured after incubations of the same SAM into progressively more concentrated IgG standard solutions, are shown in Fig. 2a. The red curve, taken as the base-line, is measured after incubation of the SAM in bare PBS solution. The subsequently measured black curve corresponds to the incubation in the 6 zM solution. Apparently the red and black curves are not significantly different. This is expected as there is a probability of 68.3% of not finding any IgG in 100 μl of a 6 zM solution. The blue curve measured at $60 \pm 30$ zM nominal concentration shows, over tens of replicates, a significant current decrease as well as a shift towards more negative gate voltages. This trend is replicated as the standard solutions with increased IgG concentration are progressively assayed, until the saturation of the response is reached. The transfer characteristics as a function of the IgG concentration are systematically reproduced by means of the SiMoT model (Supplementary Note 6) that

accurately predicts the measured curves in the whole range of IgG concentrations with the same set of few physical parameters (Supplementary Table 4). Relevantly, the SiMoT model foresees that, along with the channel current, only the threshold voltage, $V_T$, changes upon affinity binding. This is expected as the gate is designed to be about two orders of magnitude larger than the FET channel and the SAM is a high capacitance, ion-permeable membrane (Supplementary Note 3b) very responsive to electro-static changes (Supplementary Note 6). This occurrence is also corroborated by the FET response (Supplementary Note 8) being largely reduced in high saline solutions (Supplementary Fig. 12) and this was the rationale for choosing to operate the device in pure water where the Debye length, $\lambda_D$, is 100 nm.

Figure 2b shows a graphical correlation between the SiMoT $V_T$ shifts towards more negative potentials and the decrease of the gate work-function ($\phi$) at each functionalization stages and after the affinity binding[31]. The decrease in $\phi$ is generated by the electrostatic effect of a dipole directed along the $z$-axis while attached to the $x$–$y$ gate surface, pointing away from it with its positive pole[32].

The SiMoT responses (relative current variations) to IgGs in PBS standard solutions are given in Fig. 3a as squares, while the circles are the responses to IgM. An anti-Human-IgG SAM served as capturing layer and the proteins involved are all isolated from pooled normal human serum. The data points are the average over three replicates and the full line is the result of the SiMoT dose-curve modelling (Supplementary Note 7). A limit of detection (LOD) level at 11.6%, corresponding to a single protein, is estimated from the noise level of the control experiment and the high selectivity is proven by the zero response of the control assay in the whole concentration range.

The SiMoT model for the sensing dose curves, detailed in Supplementary Note 7, is based on the Poisson distribution probability suitable to account for single binding events. The model assumes that the SAM is divided into a number of domains each comprising a given number of capturing anti-IgGs. If one IgG binds to any of the anti-IgGs in a given domain, the latter supposedly entirely changes its work-function, $\phi$, due to collaborative interactions that propagate the change. The process is assumed to be irreversible and stable so as no other change in $\phi$ is possible, within that domain, if any other affinity bindings take place. The model foresees that the more compact, or electro-statically connected and defect-free the SAM is, the larger the

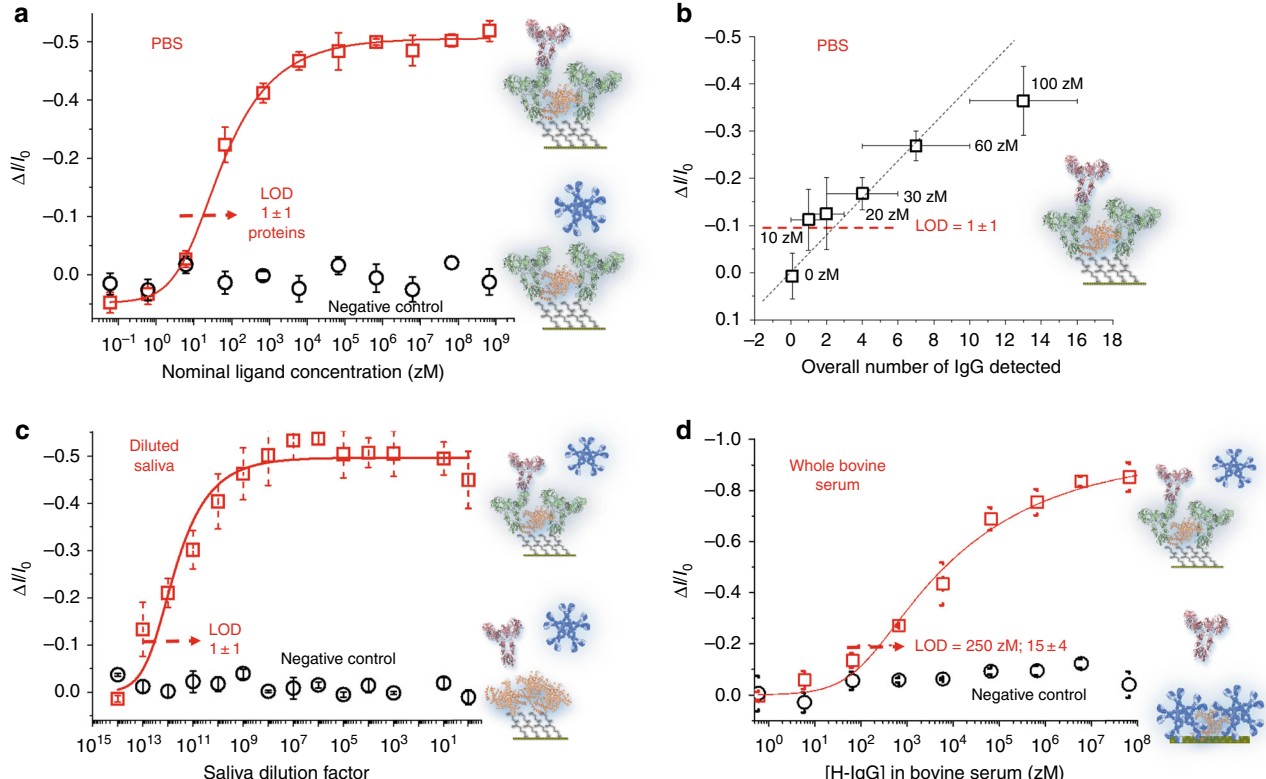

**Fig. 3** Protein detection at the physical limit in different bio-fluids. **a** The Human IgG/anti-Human-IgG affinity binding calibration curve (red squares) as the relative change of the $I_D$ current ($\Delta I/I_o$) vs. the IgG concentration. A SAM comprising both the capturing anti-IgG and the BSA is used. The black circles are the negative control responses of the anti-IgG SAM to human-IgM solutions. The proteins are assays from standard solutions in PBS. The fitting of the IgG binding curve with the SiMoT dose curve model described in Supplementary Note 7, is shown as red solid line. **b** The IgG assay in PBS carried out in the 0–100 zM range. As a single gate is used to detect molecule numbers ranging within one order of magnitude, the responses are plotted against the total number of ligands present in all the solutions sampled until the incubation at that concentration is reached, along with the relevant Poisson errors. The data provided are averages over 5 replicates and error bars are taken as one standard deviation. **c** The SiMoT is engaged in the detection of IgG in a diluted sample of real human saliva. The IgG concentration of the whole saliva, independently assayed by means of surface plasmon resonance (Supplementary Note 9), is $C_{sal} = 40 \pm 6$ nM ($10^{-9}$ M). The data points are relevant to three replicates for each dose curve while the saliva was sampled from the same batch. The reproducibility error, over three replicates, is within 4% at most. The red curve is the result of the SiMoT dose curve model carried out with the same set of two parameters used for the calibration curve in PBS. **d** The calibration curve of Human IgG spiked in whole real bovine blood serum (red squares) is shown in the 0.6 zM–6 × $10^7$ zM range. As control experiment an anti-Human-IgM SAM was used (black circles). The continuous red line is the result of the SiMoT modelling described in Supplementary Note 7

domain generated upon interaction with the first single IgG and the steeper the dose curve in the single molecule range is.

In Fig. 3b the IgG assay in PBS is zoomed into the 0–100 zM range. The response at a given concentration is plotted against the total number of ligands present in all the solutions sampled until the incubation at that concentration is performed. The relevant Poisson errors are plotted as well. As it is apparent, the responses measured at 10 zM and 20 zM (1 ± 1 particles) are beyond the LOD and prove that the SiMoT assay is capable to measure a single protein. Also the reliability seems quite high considering that the responses measured at 60 zM in Fig. 3a, b, encompassing overall the measurements carried out with eight different gates, give the same value within the error bars.

IgG is assayed at the single-protein limit also in diluted human saliva. To this end, the endogenous IgG content in a sample of whole human saliva was independently quantified to be 40 ± 6 nM (Supplementary Note 9) or $(2.4 \pm 0.4) \times 10^{12}$ molecules in 100 µl. Hence, a 1–$10^{14}$ maximum dilution of the whole saliva (in PBS) was chosen to make sure that the endogenous IgGs are absent at the single molecule level. The SAM was incubated into the progressively less-diluted saliva samples and measured following the protocol used for IgG detections in PBS standard solutions. The data, averaged over three replicates, are displayed

in Fig. 3c as squares. The red curve is the result of the SiMoT dose curve model carried out with the same set of parameters derived from the calibration dose curve in PBS (Fig. 3a). A SAM, comprising no capturing anti-IgGs but only conjugated BSAs, serves for the negative control experiment. No response was measured in this case, confirming the high selectivity of the assay. From the quantitative point of view, a very good overall agreement between the dose curves measured in PBS and in diluted saliva is found. The LOD level in the diluted saliva dose curve, based on the negative control experiment noise, is 11.3%, that corresponds to a dilution of one to $(2.79 \pm 0.13) \times 10^{12}$. This figure is in very good agreement with the total number of IgG molecules originally present in the saliva sample (independently measured as shown in Supplementary Note 9) and strikingly proves that single molecule quantitative detection is possible also in saliva, although a massive dilution needs to be undertaken to reduce the endogenous IgGs to the physical limit.

The SiMoT assay performance was tested also in a whole real bio fluid. To this end a bovine blood serum added (spiked) with Human IgGs was analysed. In principle, the endogenous Bovine IgGs should not selectively bind to anti-Human-IgGs. However, it is not obvious that this holds true at the single-molecule level also considering that $10^{12}$ endogenous bovine IgGs should be present

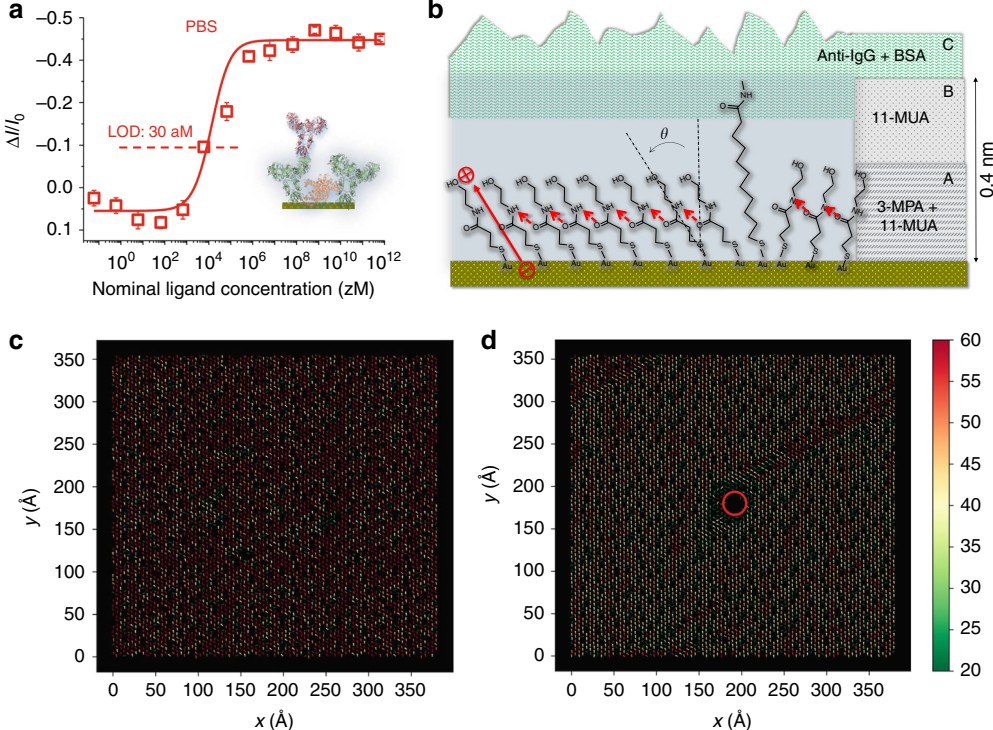

**Fig. 4** The chem-SAM work function change. In (**a**) the dose–response curve of the IgG assay measured with the gate covered by the sole bio-SAM directly physisorbed on the gold surface, is given. The fitting of the IgG binding curve with Supplementary Eq. 20, derived in Supplementary Note 7, is shown as red solid line. In (**b**) detailed illustration of the chem-SAM is provided, with the H-Bonds visualized as red dashed arrows. The angle **θ** is defined by the vector originating from the sulfur atom and pointing towards the oxydril oxygen atom of activated and blocked 3-MPA chain and the z-axis normal to the gate surface plane and aligned with the gating field. In (**c**) the direction and the occupancy (%) of the H-bonds resulting from the analysis of MD trajectories are shown under the gating-field. The color codes indicate the percentage of frames in which the H-bond is established. In (**d**) the simulation is carried out on the same system but a defect mimicking the impact of the affinity binding event is simulated by imposing as coordinates of the central circular region (red circle) a disordered conformation (stably missing of H-bonds). See Supplementary Note 11 for details

in 100 μl of bovine serum. As control experiment an anti-Human-IgM SAM was used instead of the sole BSA as the latter is expected to be less effective in minimizing non-specific binding in the bovine serum. The data provided in Fig. 3d show the dose curve for the spiked Human IgG (squares) over a range from 0.6 zM to $6 \times 10^7$ zM along with the control experiment (circles). A LOD of 250 zM, corresponding to $15 \pm 4$ molecules, is evaluated in this case probably due to non-specific binding. The non-specific adsorption on the SAM surface of a layer that is insensitive to affinity-bindings, likely adds a parasitic low capacitance that modulates the FET response mitigating the electrostatic high sensitivity of the SAM. Maybe just a 1–3 dilution is enough to reduce this effect. Still, even in this challenging biological environment, the LOD measured is a world record in label-free protein detection with a non-nanometric device, being more than two orders of magnitude better than the state-of-the-art[25].

**The role of the chem-SAM.** In Fig. 4a the IgG dose curve in PBS is measured, in the so far explored concentration range, with a gate covered by the bio-SAM. Meaning that in this case there is no chem-SAM attached to the gold and the bio-SAM is directly physisorbed on gold. The configuration results in transistor sensor with a LOD as high as 30 aM. Such a value, comparable to the state-of-the-art[25], is three orders of magnitude higher than the one necessary for single molecule detection in 100 μl. This proves the pivotal role of the chem-SAM to reach the single molecule detection limit. The comparison between the extracted dispersion functions (Supplementary Fig. 11) for the dose curves

in Fig. 3a and Fig. 4a further supports this point. Indeed, the SiMoT dose curve modelling shows that the SAM comprising both the chemical and the biological component (Fig. 3a), offers a probability larger than 99% to find domains including as many as $10^{11}$ anti-IgG. According to the model that we are suggesting, in these few domains all the anti-IgGs, actually the chains of the chem-SAM under their footprint (vide infra), undergo a work-function, $\phi$, change that, altogether, generates the amplification needed to measure the single binding event. It is also apparent that, when the chem-SAM is not included (Fig. 4a), the bio-SAM alone offers, with a probability of 96%, domains that are two orders of magnitude smaller being composed of $10^9$ anti-IgGs. Thus, the sole bio-SAM is probably either too defective or it lacks a sufficiently large network of cooperative electrostatic interactions. Hence, the amplification effect that enables the single molecule sensitivity is to be sought in the chem-SAM layer.

The chem-SAM structure is detailed in Fig. 4b where the more compact shorter chains along with the free to orient longer ones, are visible. The chains bear a dipole moment (red arrow Fig. 4b) oriented along their axis with the positive pole pointing away from the gold surface. This orientation is largely determined, particularly in the shorter chains, by the strong, specific[28] and oriented[33] Au–S bond. The chains' dipole moment orientation on gold, suggested also by the computational study (Supplementary Note 11 and Supplementary Fig. 15), lowers the vacuum level and hence the gold work function $\phi$, resulting in the measured $V_T \approx -0.2$ V shift (Fig. 2b), as a p-type OSC is used[34].

The amide group resulting from the blocking reaction with ethanolamine originates an H-bond whose associated dipole

moment is oriented from the oxygen of the amide group in one chain and points towards the hydrogen of the amide group of a neighbouring one (Fig. 4b). Given the rather regular assembly of the activated-and-blocked 3-MPA in the chem-SAM, an H-bonding network likely forms. This picture is consistent with recently published works on H-bonding networks in similar SAMs[29,35]. The 11-MUAs, that are too far apart to engage into an H-bond, can introduce a localized weak disorder in the H-bonding network. These features are derived from the analysis of the trajectory resulting from the molecular dynamics (MD) simulation of the chem-SAM in the gating field (Fig. 4c) where the direction and the occupancy of the detected H-bonds are shown. The colour code indicates the percentage of frames in which the H-bond is established (Supplementary Note 11). MD simulations exploit an implicit solvent model for water as such an approach allows us to properly simulate the system in the absence (dielectric constant $\varepsilon = 80$) and in the presence of an electric field that generates a charge double layer (dielectric constant $\varepsilon = 6$).

An anti-IgG footprint occupies a surface of ~100 nm² covering approximately 400 chains (each ~5 Å wide) of the chem-SAM[36]. This is confirmed considering the measured number of capturing anti-IgGs in the SAM covering the 0.6 cm² wide gate, evaluated by surface plasmon resonance to be $(5.92 \pm 0.3) \times 10^{11}$ (Supplementary Note 1). This figure compared to the number of chains on the whole gate area ($3 \times 10^{14}$ 3-MPA and $3 \times 10^{13}$ 11-MUA), estimated from the reported full coverage density[28,36], returns a single anti-IgG over 400–500 chains ratio. It is then reasonable to assume that the covalent binding of an anti-IgG occurs mostly at one of the 40–50 11-MUAs present under its footprint area, as they are longer and more flexible than the 3-MPAs.

**Description of a possible sensing mechanism**. The IgG binds to the anti-IgG Fab fragment with its fragment crystallisable (Fc) region thus, for the affinity binding to take place, the capturing anti-IgG needs to expose at least one of its Fab fragments to the solution. The SAM surface characterization (Supplementary Note 3), its thickness (Supplementary Note 2) but most compellingly, the measured availability of ~10¹² bio-active capturing anti-IgGs (Supplementary Note 1), strongly suggest that the anti-IgGs packed in the SAM largely lay edge-on with one of their Fab fragments pointing away from the gate surface. The resulting layer is probably characterized by a certain degree of order, albeit a very low one. A simplified picture of the SAM structure highlighting these features is provided in Fig. 1c.

It is reported that, at physiological pH, antibodies bear a dipole moment oriented from the Fc to the Fab region (Fig. 1d)[37,38]. In the paper by Emaminejad et al[37]. the antibodies are deposited from a PBS solution so the dipole moment includes the counter ions. Hence we assumed that the dipole moments, associated with the anti-IgGs laying on the gate, have a component that points away from the substrate as well. Such overall weakly oriented dipoles can endow the bio-SAM of limited electrostatic properties. It can also possibly add a weak electrostatic driving force component to the affinity binding process.

It is acknowledged that antigen/antibody binding reactions are exothermic[39,40]. The energy involved, measured in a single IgG/anti-IgG binding experiment, is of the order of tens of kJ/mol[41]. The allosteric nature of the antigen/antibody binding[42] foresees that the large conformational change occurring in the Fab fragment regions of the capturing antibody upon binding, is transferred to its Fc region[43]. Given the partial orientation of the capturing anti-IgGs in the SAM previously discussed, the Fc region should be bound to an 11-MUA chain. Hence the major conformational change transferred to the Fc region, likely impacts on this chain as well as on the chem-SAM around it. We therefore

assume that the binding energy, at least partially, can be transferred from the anti-IgG/IgG complex to the chem-SAM.

The 3-MPA and the 11-MUA chains desorb as disulphides[44]. This process requires an estimated energy of 100 kJ/mol for a generic alkanethiols pair[45]. Although the exact estimate of the energy needed to desorb an activated-and-blocked 3-MPA pair in the chem-SAM is not known, it could be assumed that the process involving the loss of two chains starts at the expenses of the energy released during the affinity binding. The subsequent exposure of the chem-SAM to a reductive potential of −0.7 V, occurring during the measurement of the FET current, can accomplish their full desorption, thus possibly generating an irreversible defective region[46]. Indeed a massive desorption involving a large number of chains does not occur, as the maximum potential reached during the FET current measurement is kept lower than the electrochemical potential of the reductive reaction[47]. The described loss of chains can hence account for the origination of a defect in the chem-SAM, along with the associated irreversible disorder.

A possible explanation on how an extremely small defective region generated in the chem-SAM can trigger a work function change over an area orders of magnitude wider, once the SAM is immersed in an electric field, can be provided by means of the computation study detailed in Supplementary Note 11. To this end, MD simulations were performed on a cell formed by 5892 activated-and-blocked 3-MPA and 589 activated-and-blocked 11-MUA chains to reproduce the real chem-SAM structure. An overall area of $1.45 \times 10^3$ nm² is covered by the modelled system. A uniform electric field reproducing the gating field on the chem-SAM is applied, while the affinity binding is simulated by restraining the coordinates of a 4.52 nm² region ($3 \times 10^{-3}$ of the whole area) in a disordered conformation (permanent loss of H-bonds). The results of the simulations performed in the absence and in the presence of the defect region, are shown in Fig. 4c and d, respectively. The defect area is highlighted with a red circle in Fig. 4d. Larger versions of these figures are provided as Supplementary Fig. 18 and Supplementary Fig. 19. The direction and the occupancy of the H-bonds are shown under the gating-field and the H-bonds are visualized with arrows while the color codes indicate the percentage of frames in which the H-bond is established. Apparently, the imposed defect in a $3 \times 10^{-3}$ portion of the whole area, can generate a completely different pattern of the H-bonding network and a change of the gate work function can be also inferred. Moreover, being the simulated chem-SAM free from defects (other than the one generated by the binding), the extension of the domain where the work function is changed covers the whole simulated area.

The averaged values of the $\theta$ angle between the shorter chains' backbone and the gating-field direction (Fig. 4b) are also provided by the analysis of the MD trajectories and quantify the changes of the direction of the dipole moments associated with the chains and, eventually, the associated changes of the gate work function. The $\theta$ values, in the gating field and upon the single molecule binding, are given in Supplementary Table 5. The simulations show that, in the absence of any binding, $\theta$ increases by 22% when the gating electric field is applied. This is indicative of the strengthening of the H-bonds in the electric field that causes the shorter chains to further bend giving rise to a strained network configuration. The nematic, $P_2$, and the dynamic, $S$, order parameters[48] have been computed (Supplementary Tab. 5) to assess the degree of order and the stability of the H-bonding network. $P_2$ values in the presence of the field are higher by 93% indicating that the presence of the applied electric field induces a higher order in the system. S, varying from negative to positive values as the order stability is improved, is negative only in the absence of the electric field. Hence, the electric field strains, orders and stabilizes the H-bonding network in the chem-SAM.

When the binding occurs and no field is present, $\theta$ does not change, while, when the field is switched-on a $\theta$ decrease of 16 % is computed. This proves that the field is necessary to activate the electrostatic collective interactions that enable the measurement of a single-binding event. Moreover, a decrease in $\theta$ implies a decrease of the gate work function. This is in agreement with the experimentally measured direction of the threshold voltage shift towards more negative potentials with the binding.

Altogether the performed simulations can give a possible explanation on how the disordered area generates, in the absence of an electric field, a change in the gate work function that, being restricted to a nanometric-wide region, cannot be detected by the millimetre-size gate of a FET. When the field is switched-on the H-bonding network sets into a strained configuration endowed with higher order and stability that enable the propagation of a work function change thus making the signal measurable by the FET. More in details, when few H-bonds are (permanently) removed in the defective area, the neighbouring chains can more easily align their dipole moments along the field direction (smaller $\theta$) and this generates a perturbation in the strained network that eventually adjusts to compensate for the local change. The change in the pattern is shown in Fig. 4b. Such an adjustment involves the propagation of a $\theta$ decrease through the collective electrostatic interactions of the network. The degree of order and the stability of the network is high also after the binding as proven by the orders parameters values (Supplementary Table 5). As the H-bonding network rearrangement after the binding is sustained by the field, it can propagate until a pre-existing defect in the SAM, such as a domain wall, stops the process.

## Discussion

This work represents the first successful attempt to conceptualize cells' ability to detect ligands at the single-molecule level by means of a mm-size FET-transducer. The label-free SiMoT platform is enabled by a FET comprising a SAM of highly packed capturing antibodies covalently attached to the gold gate surface that make a single binding event cross section sufficiently high. The SAM is characterized by a hydrogen-bonding network that is assumed to create collective electrostatic interactions and can contribute to define the gate work function. A possible explanation of the extremely high sensitivity involves an amplification that is achieved via the propagation of the gate work function change upon affinity binding that is driven by the elicited electrostatic cooperative interactions that are sustained by the gating electric field. Hence the combination of an electrostatically connected layer and the presence of the electric field could enable the amplification process to take place after the single binding, thereby making the measurement of a single binding event possible. The MD simulations give some indications that the propagation can cover an area orders of magnitude larger than the defect generated in the network by the binding. Such a spreading is supposedly stopped only by defects, such as domain walls, pre-existing in the SAM layer. Hence, the less defects are present in the SAM the more one single binding event will provoke a large work function variation with the associated large measurable signal the dose curve response is steeper.

The resulting work-function change is converted and further amplified in a reading current by the SiMoT operated in water. This study paves the way to new fundamental investigations in the field of surface-confined antibodies/proteins electrostatic and cooperative interactions, starting a novel front in the area of interfacial soft-matter interfaces and bioelectronics.

From a more applicative perspective, the pivotal aspect of the present study is the functionalization of the gate electrode

combined with the EGOFET architecture. The proposed SiMoT could be implemented in several technology platforms based for example on organic and polymeric materials, graphene, metal dichalcogenide as well as amorphous oxides transistors. Moreover, given the general approach engaged in the bio-functionalization procedure, the platform is in principle applicable for the detection of a wide spectrum of bio-markers. Indeed, detection to the ultimate limit has been achieved as proves of principles also with IgM binding to anti-IgM and the C-reactive protein binding to anti-CRP. Further on it has been also proven for DNA and peptide single-molecule detection.

Prospectively, the proven single-molecule sensitivity can be an advantage per-se, for instance in clinical application where a single bio-marker is already symptomatic of a progressive pathology. Moreover, the reduced dynamic range resulting from the φ-change measured with more defect-free SAMs, can be a plus in designing a sensor that acts as a binary device capable to spot, with an on-off-type response, the presence of a single bio-marker in a fast and reliable fashion. Likewise, this device could be useful to assess the absence of a given protein or bio-marker at the single-molecule level in a bio-fluid. On the other hand, the dynamic range can be increased by two orders of magnitude by using a gate that has pre-defined isolated domains that limit the propagation of the φ-change for each one single-event (L. Torsi et al. "A field-effect transistor sensor." European Patent Application no. 17177349.2, filed 2017). Hence, the proposed platform opens up new relevant opportunities and associated markets in the field of low-cost array platforms (L. Torsi et al." A field effect transistor sensor and the corresponding array." PCT Patent Application no. IB2018/050491 filed 2018.) for point-of-care early-diagnostic with a foreseen huge impact of a simple, yet widely applicable, ultra-sensitive immunoassay technology.

## Methods

**Materials.** Poly(3-hexylthiophene-2,5-diyl), P3HT (Sigma-Aldrich, regioregularity > 99%), with an average molecular weight of 17.5 kDa (g mol$^{-1}$), was used as semiconductor with no further purification. 3-mercaptopropionic acid (3-MPA), 11-mercaptoundecanoic acid (11-MUA), 1-ethyl-3-(3-dimethylaminopropyl)-carbodiimide (EDC), N-hydroxysulfosuccinimide sodium salt (sulfo-NHS) and K$_4$[Fe (CN)$_6$]·3H$_2$O (98.5%) were purchased from Sigma–Aldrich and used with no further purification. The anti-Human Immunoglobulin G (anti-IgG), is a Fc-specific antibody affinity produced mostly in goat (molecular weight ~144 kDa) while the human IgG (~150 kDa) affinity ligand and the human IgM (~950 kDa) ligand, were extracted from human serum. All the immunoglobulins used are polyclonal antibodies. Bovine serum albumin (BSA) has a molecular weight of 66 kDa. Both the capturing-antibody and the ligands as well as ethanolamine and BSA were purchased from Sigma–Aldrich and readily used. Water (HPLC-grade, Sigma-Aldrich), potassium chloride (Fluka, puriss p.a.) and ethanol grade, puriss. p.a. assay, ≥ 99.8 %, were used with no further purification.

**Electrolyte gated (EG)-OFET fabrication.** Electron-beam evaporated gold source (S) and drain (D) interdigitated electrodes (50 nm, thick) were photo-lithographically defined on a Si/SiO$_2$ substrate. A prior deposited layer of titanium (5 nm) served as adhesion layer. The distance between two fingers defines the channel length ($L = 5$ μm), while the perimeter of all the equipotential fingers is the channel width ($W = 1280$ μm). The transistor channel area covered by the OSC was $6.4 \times 10^{-3}$ cm$^2$. Prior to the electrodes patterning, the SiO$_2$ surface was cleaned in an ultrasonic bath of solvents of increasing polarity (acetone and iso-propanol respectively) for 10 min each. After the S and D electrodes definition, a P3HT solution (2.6 mg ml$^{-1}$ in 1,2-dichlorobenzene) filtered with 0.2 μm filter was spin-coated at $2 \times 10^3$ r.p.m. for 20 s and annealed at 80 °C for 1 h. The P3HT film showed a hydrophobic surface characterized by a contact angle of 103 ± 3°. A polydimethylsiloxane well was glued across the interdigitated channels area and was filled with 300 μl of water (HPLC-grade) acting as gating medium. A Kapton® foil (area of ~0.6 cm$^2$) covered by e-beam evaporated gold (50 nm) on titanium (5 nm) served as the gate (G) electrodes. The gate was stably positioned on the water on top of the well in correspondence of the electrodes interdigitated area.

**Gate bio-functionalization protocol.** Before functionalization, the gold surface was cleaned in an ultrasonic bath of isopropanol for 10 min and UV/ozone cleaned for 10 min. The chemical SAM (chem-SAM) on the gold surface comprises a layer of mixed alkanethiols terminating with carboxylic functionalities. To this end, a 10

mM solution consisting of 10:1 molar ratio of 3-MPA to 11-MUA was prepared in ethanol. The cleaned gold surface was immersed in the 3-MPA and 11-MUA solution and kept in the dark under constant $N_2$ flux for 18 h at 22 °C[30]. The carboxylic groups were activated afterwards in a 200 mM EDC and 50 mM sulfo-NHS aqueous solution for 2 h at 25 °C. This process is addressed in the text as EDC/sulfo-NHS chemical activation or simply as activation. The IgG capturing SAM was generated, subsequently, through conjugation between the amine groups of the antibodies and the activated carboxylic groups on the gate surface, by immersing the gate in an anti-IgG phosphate buffer saline (PBS) solution for 2 h at 25 °C. The solution was composed of 0.7 µm (0.1 mg ml$^{-1}$) of anti-IgG and 10 mM (KCl 2.7 mM and 137 mM NaCl) of PBS at a pH of 7.4 and an ionic-strength (i$_s$) of 162 mM. Afterwards, to saturate the unreacted sulpho-NHS groups, the anti-IgG SAM was further treated with ethanolamine 1 M in PBS 10 mM for 1 h at 25 °C. This latter step is addressed as the surface "chemical-blocking". Finally, the bio-functionalized gate was immersed in a 1.5 µm (0.1 mg ml$^{-1}$) BSA solution in PBS 10 mM for 1 h at 25 °C. This step of BSA physisorption is addressed as the "bio-blocking" of the gate surface. Both the conjugated anti-IgG and the adsorbed BSA form the layer addressed as "bio-SAM". The gate functionalized with both the chem-SAM and the bio-SAM is addressed in the text as the "SAM" and is schematically depicted in Fig. 1c. For the control experiment in Fig. 3c the bio-SAM is formed by conjugating BSA (instead of anti-IgG), followed by the surface bio-blocking with BSA. The physical adsorbed SAM, used for the assay in Fig. 4a, was deposited by skipping the chem-SAM conjugation, hence depositing anti-IgGs and BSAs, by immersing the gate in the elicited anti-IgG PBS solution for 2 h at 25 °C and subsequently into the BSA solution in PBS for 1 h at 25 °C. After each step of the functionalization protocol, the gate was rinsed thoroughly in water to remove possible residues.

**Sensing measurements.** The SiMoT electronic curves were measured by positioning the gold-gate electrode in contact with the water which functions as electrolyte gating medium (Fig. 1a, b). The FETs current–voltage curves were measured with a semiconductor parameter analyser equipped with a probe station, in air and at RT (20–22 °C). The FETs were tested in the common-source configuration. As customary, for the output characteristics, the drain current ($I_D$) was measured as a function of the drain voltage, $V_D$, at different gate voltages, $V_G$, the latter ranging between 0 and −0.5 V, in steps of −0.1 V. Typical output curves are shown in Supplementary Fig. 7a. For the transfer characteristics $I_D$ was measured as a function of $V_G$ (ranging from −0.1 to −0.7 V in steps of −0.01 V) at a constant drain voltage of −0.4 V (Supplementary Fig. 7.1b). The voltage ranges were tuned to minimize the gate current ($I_G$) associated with electrochemical processes that could produce massive reductive desorption of the chem-SAM thiolate chains as well as the oxidative degradation of the SAM. To control and minimize such processes, $I_G$ was always acquired along with $I_D$ and all the curves were measured in the forward and reverse mode to evidence the occurrence of hysteresis. A typical $I_G$ curve showing current values three orders of magnitude lower than $I_D$, very limited faradaic-currents and a low hysteresis, is shown in the inset of figure Supplementary Fig. 7b. Before proceeding with the sensing measurements, $I_D$ was stabilized by cycling the measurement of the transfer curve of the P3HT FET comprising a clean bare gold gate, until the last three current traces overlapped. During this process, the low-mobility trap states of the P3HT OSC were filled, leading to a stable $V_T$ value. Once the P3HT $V_T$, or equivalently its work-function, was stabilized, further measured $V_T$ shifts only relate to changes occurring in the gate work-function, $\phi$. After the OFET stabilization was accomplished, a transfer curve (black curve in Supplementary Fig. 7b) was recorded and stored as reference current level. A SAM gate was then incubated (at RT and in the dark) for 10 min in 100 µl of PBS. The gate was removed from the PBS solution, washed thoroughly with HPLC water, mounted on the FET (exactly replacing the bare gold-gate previously used) and a new transfer characteristic was recorded, after stabilization through cycling was performed. This latter process was carried out to stabilize the work-function of gate in the water environment. The red curve shown in Supplementary Fig. 7b and in Fig. 2a, is a typical stabilized current measured with the SAM gate exposed to PBS, addressed as the "$I_0$" current base-line. After the measurement of the $I_0$ base line, the same SAM gate was immersed and incubated (at RT and in the dark) for 10 min in 100 µl of the PBS standard solutions of the ligands (IgG or IgM) with nominal concentrations ranging from $6 \times 10^{-2}$ zM to $6.67 \times 10^8$ zM. The evaluation of the error on the concentrations, or equivalently on the number of proteins sampled in 100 µl, was carried out considering both the Poisson and the dilution error. The total error at each concentration is computed as the square-root of the sum of the squares of the dilution and Poisson's uncertainties, the latter expressed also in molarity (Supplementary Table 3). All the errors are taken as one standard deviation. The incubation time was estimated to allow enough time for the IgG or IgM proteins present in the incubating solution to impinge on the anti-IgG layer. A diffusion constant as high as $4 \times 10^{-11}$ m$^2$ s$^{-1}$ was measured for IgG in PBS solution, resulting in a root mean square displacement (RMSD) explored in 10 min, of 0.4 mm. Considering the incubation volume being as low as 100 µl, the RMSD is high enough to reasonably assume that even few proteins can impinge on the gate surface during the incubation. This is even more true, considering that dipole moments are associated both with antibodies and ligands that likely enables also for electrostatic driven attractive forces to be in place. After incubation in each of the PBS standard-solutions of IgG or IgM

starting from the more diluted one, the SAM was washed thoroughly with PBS first and then with water to remove the unreacted ligands away and a further I–V transfer curves were measured. Also in this case extra care was taken to positioning the gate always at the same height (ca. 4 mm) as for the measurement of the base-line, $I_0$. Also at this stage a stabilization of the current was carried out by measuring the transfer curve, in the forward and reverse run, until a few stable traces were seen. The stabilized currents measured after incubation in each standard solution are addressed as the "I" signal at a given concentration. The $\Delta I/I_0 = [(I − I_0) / I_0]$ is the normalized electronic response at a given concentration and the relevant dose–response or calibration curve is obtained by plotting these data points, at the $V_G$ value that maximizes the trans-conductance $\delta I_D/\delta V_G$ (falling generally in the −0.3 V to −0.4 V range), at all the investigated concentrations. One sole SAM gate (measured also with the same P3HT channel) was used to measure a whole IgG (or IgM) dose–response curve spanning eleven orders of magnitude ($6 \times 10^{-2}$ zM–$6.67 \times 10^8$ zM). All the data points are averaged over three replicates with each full calibration curve being measured on a different SAM. The resulting reproducibility error is computed as one standard deviation. Further three SAM gates were engaged to measure the IgM negative control dose curves and data are presented as averages along with one standard deviation error bars. After measuring each whole calibration curve, such as the ones shown in Fig. 2d, the bare gold gate previously used to measure the reference current level (as for instance the black curve in Supplementary Fig. 7b) was positioned back on the FET comprising the same P3HT device, and a final transfer curve is recoded. The resulting curve, is shown as the blue trace in Supplementary Fig. 7b (Supplementary Note 4). It is clear that it almost overlaps to the reference level previously stored (black curve). Indeed, the relative variation of the current measured on the bare Au-gate serving as control, before and after the measurement of the whole dose–response curve, was considered acceptable only when falling within 5–8%. This error level was used to validate the data set of one calibration curve as it proved that the current changes measured were due to the biochemical interactions and not to spurious effects such as a P3HT degradation. As a further control experiment a SAM gate was incubated (at RT and in the dark) for 10 min in 100 µl of PBS. This current was measured eleven times after consecutive incubations, for 10 min in PBS, of the same SAM gate. The relative changes of each trace, as compared to the first one, are shown in Supplementary Fig. 14 (Supplementary Note 10), to be as low as 1.2 ± 0.1%.

**Computing the limit-of-detection.** The LOD was customarily computed as the concentration providing a response equal to the average of the noise level plus three times the noise standard deviation. The relative current variation in the IgM negative control assay in PBS (Fig. 3a), taken as the level-of-noise, is (2.0 ± 2.3)%. The computed LOD response level for the IgG assay in PBS (Fig. 3a) is hence 11.6% resulting in a LOD of 10 zM approximated, in 100 µl, to as few as 1 ± 1 ligands. The LOD value was evaluated also for the IgG assay in saliva to be 11.3% (Fig. 3c). The LOD increases to 250 zM (15 ± 4 particles) for the assay in whole bovine serum (Fig. 3d). Here the level-of-noise, is (5 ± 5)% hence the computed LOD response level for is 20%.

**Detection in saliva.** Saliva samples were collected by passive drool method from a healthy human female volunteer. The sampled saliva was divided into aliquots of 500 µl and frozen at −20 °C immediately after collection. When needed, a saliva aliquot was brought to room temperature, vortex and centrifuged at $1.5 \times 10^3 \times g$ for 15 min. The supernatant was collected and progressively diluted 1:10$^{15}$ times in PBS. For the assay, a single SAM gate was incubated for 10 min in each diluted saliva solution, starting from the more diluted to more concentrated one, washed thoroughly in PBS and with HPLC water and measured with the SiMoT. As negative control experiment a gate functionalized with the sole BSA (both conjugated and adsorbed), was incubated in saliva solutions diluted in PBS.

**Detection in whole bovine serum.** Also in this case the sensing was carried out by incubating, the gate bio-functionalized with the SAM comprising the anti-Human-IgG capturing proteins (at RT and in the dark) for 10 min in 100 µl of whole bovine serum. The same SAM is then immersed and incubated in the bovine serum spiked with standard-aliquots of Human-IgG in PBS. The maximum volume of PBS added to the whole bovine serum was 0.2%. The resulting nominal concentrations of the Human-IgG in the bovine serum ranged from $6 \times 10^{-1}$ zM ($10^{-21}$ M) to $6 \times 67 \times 10^7$ zM.

**Computational details.** In order to model a realistic SAM surface, a computational protocol inspired by Ref. 49, herein described, was adopted. All the periodic density functional theory (DFT) calculations were performed using the CP2K/Quickstep package[50]. In particular, a hybrid Gaussian and plane-wave method was used. All calculations were performed at the PBE/TZVP level of theory[51] using GTH pseudopotentials[52] together with a 400 Ry plane wave cut-off. Dispersion forces were taken into account with the Grimme DFT-D3 method[53]. Molecular dynamics (MD) simulations were performed using the NAMD 2.12 package[54] and the CGenFF[55,56] force-field with RESP charges[57] calculated at the HF/6–31 G(d,p) level of theory using the Gaussian package[58]. To reconstruct the gold surface, a bulk-gold supercell was optimized by means of DFT, obtaining the cell parameter $a =$

4.152 Å, close to the experimental one[59], $a_{exp} = 4.078$ Å. The 111 Au surface was cut from this supercell and relaxed keeping the gold atoms belonging to the lowest layer fixed so as to simulate bulk constraints (Supplementary Fig.6). Finally, eighteen 3-MPA EDC/sulfo-NHS chemical activated and blocked with ethanolamine were added to the system following the $(\sqrt{3} \times \sqrt{3})$R30° configuration[60–62]. The resulting system was optimized at the PBE/TZVP level of theory. The DFT optimized configuration was replicated to form a cell 396 Å × 366 Å wide in the $x$–$y$ plane, comprising 6480 activated and blocked 3-MPA molecules. To reproduce the chem-SAM comprising 3-MPA and 11-MUA (10:1) 589 3-MPA chains were replaced by EDC/sulfo-NHS chemical activated and blocked with ethanolamine 11-MUA. The gold atoms were removed and, to ensure the charge neutrality of the system, hydrogen atoms were added and kept constant during the MD simulations to mimic the bulk constraints. A cut-off of 12 Å was applied to the Lennard–Jones interactions employing a switching function (switching radius of 10 Å). Electrostatic interactions were treated using the Particle–Mesh–Ewald (PME) method[63], with a real-space cut-off of 12 Å and a grid spacing of 1 Å per grid point in each dimension. All simulations were performed in the canonical ensemble with the Langevin thermostat[64] (damping coefficient 1 ps$^{-1}$). A time step of 1 fs was used, storing the coordinates every 10,000 steps (10 ps). The simulation cell, depicted in Supplementary Fig. 17, was used to carry out four different 100 ns-long MD simulations at $T = 25$ °C under the following different conditions: (i) in implicit solvent (water) setting the dielectric constant $\varepsilon = 80$; (ii) in same condition as (i) but in the presence of a defect generated by restraining the coordinates of a region (force constant $k = 1$ kcal mol$^{-1}$ Å$^{-2}$) with a radius of 12 Å in a disordered conformation (permanent loss of the H-bonds); (iii) as in (i) but in the presence of an uniformly applied electric field ($E = -0.1$ V/nm, reproducing the gate field in the SAM) oriented in the $z$-axis direction perpendicular to the SAM surface and setting the dielectric constant $\varepsilon = 6$ to account for the charge double layer; (iv) as in (ii) but in the presence of the elicited uniformly applied electric field and setting the dielectric constant $\varepsilon = 6$ to account for the charge double layer. The H-bond occupancy was computed using as thresholds a distance atom acceptor (AA)—atom donor (AD) equal to 3 Å and an angle AD–H–AA equal to 150°. Moreover, the orientational order of the simulated systems was investigated by computing the nematic order parameter $P_2$ and the dynamic order parameter $S$[48,65,66] (see Supplementary Information for methodological details).

**Data availability**. The authors declare that the data supporting the findings of this study are available within the paper and its supplementary information files.

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

## Acknowledgements

We acknowledge A. Tiwari for collaborating to the gathering of some of the earliest detections at 60 zM not included in this work. We thank M.V. Santacroce, R. Osterbacka, H. Harma and F. Palmisano for useful discussions. We thank Paolo Romele for his relevant contribution to the EIS measurements and Rosa Filazzola for revising the English form. R. Sciorsci is acknowledged for providing the Bovine Serum. A. Notargiacomo is acknowledged for providing the S&D interdigitated mask. "OrgBIO" Organic Bioelectronics (PITN-GA-2013-607896), "Sense-of-Care" OFET biosensors for point-of-care applications (PITN-2012-GA-316845), PON SISTEMA (MIUR), Future in Research "FLOW" Dispositivi EGOFET flessibili a bassa tensione per la sicurezza in campo alimentare (Codice Pratica: ML5BJ85) projects and CSGI are acknowledged for partial financial support. We acknowledge RECAS and the CINECA awards no. HP10CJSIFE-AQP4-NT and no. HP10BGY23X-OAP-AQP4 under the ISCRA initiative for the availability of high-performance computing resources.

## Author contributions

E.M. fabricated the SiMoT and performed the sensing measurements. K.M. and B.H. characterized the gate by electrochemical and surface plasmon resonance measurements. E.M., B.H. and K.M. worked under the supervision of L.T.; C.D.F. fabricated the interdigitated contacts; M.G. performed the modelling of the transistor transfer curves under the supervision of F.T.; F.T. supervised and analysed the EIS measurements. G.F.M. and D.A. performed DFT calculations and MD simulations; G.P. produced the SiMoT dose curve model; G.S. performed the morphological surface analysis and contributed to the explanation of the sensing mechanism. L.T. conceived the idea of the SiMoT device, contributed to the understanding of the sensing mechanism and wrote the manuscript that was revised and approved by all the authors.

## Additional information

**Competing interests:** The authors declare no competing interests.

