## [Peer Review File · Nature Communications]

REVIEWERS' COMMENTS:

Reviewer #1 (Remarks to the Author):

I went through the revised manuscript. Presenting the explanation as tentative is an essential improvement and I recommend publication in Nature Communications. There are however two specific issues raised in my last report to Nature that was not addressed at all:

1. MD simulations exploit a highly simplified models for water, that is included implicitly only through a dielectric constant acting on the electrostatic of the system. This is a rough approximation; in the presence of explicit waters there would be H-bonds between water and the SAM that may well change the H-bond network shown in Fig.4(c,d). This issue should be clearly stated in the main text, to place the simulation results in the proper perspective.

2. In ref. 42 I do not find the value of 70 ± 21 kJ/mol for the energy released upon binding of IgG with anti-IgG (stated on p.10 in the manuscript). Actually in that work I could find values only for activation barriers (for unbinding), that of course have nothing to do with released energies. Please clarify.

Reviewer #2 (Remarks to the Author):

I have refereed this paper twice for Nature and so am familiar with it and I assume the editors have my previous reports. I feel the work is interesting, novel and thoroughly done. It will attract a lot of interest. I am unsure about the mechanisms but the authors have addressed all the experimental queries I had that could contradict their claims of sensitivity in complex samples. THE only changes they did not make from this version compared to the last was to do with language, being more tentative regarding the mechanism and using slightly simpler, more understandable expressions. I also note the addition of the SiMoT abbreviation which I think is silly but so be it.

I am happy for this paper to be accepted.

Reviewer #3 (Remarks to the Author):

The remaining minor points from my previous reviews were addressed. I am in favor of publication of this version.

We have revised the manuscript according to Reviewers' comments and in the following the reply to the issues is provided:

Reviewer #1 (Remarks to the Author):

I went through the revised manuscript. Presenting the explanation as tentative is an essential improvement and I recommend publication in Nature Communications. There are however two specific issues raised in my last report to Nature that was not addressed at all:

1. MD simulations exploit a highly simplified models for water, that is included implicitly only through a dielectric constant acting on the electrostatic of the system. This is a rough approximation; in the presence of explicit waters there would be H-bonds between water and the SAM that may well change the H-bond network shown in Fig.4(c,d). This issue should be clearly stated in the main text, to place the simulation results in the proper perspective.
2. In ref. 42 I do not find the value of 70 ± 21 kJ/mol for the energy released upon binding of IgG with anti-IgG (stated on p.10 in the manuscript). Actually in that work I could find values only for activation barriers (for unbinding), that of course have nothing to do with released energies. Please clarify.

Reply to Reviewer # 1:

We apologize for overlooking these issues in the previous report. Here are the replies:

1- To comply with to Reviewer #1 request, we have introduced at the end of the first paragraph at the top of pag. 9 the following sentence: "MD simulations exploit an implicit solvent model for water as such an approach allows us to properly simulate the system in the absence (dielectric constant $\epsilon = 80$) and in the presence of an electric field that generates a charge double layer (dielectric constant $\epsilon = 6$)".

2- We thanks the reviewer for pointing this out. Indeed, in the cited paper the estimated barrier energies in the binding process are given in $K_B T$. We used this value to provide an estimate of the energy involved in a single IgG / anti-IgG binding. However, following the Reviewer comment in the new version we decided to provide this value just as an order of magnitude estimate, hence the following sentence is now present in the manuscript at pag. 10: "The energy involved, measured in a single IgG/anti-IgG binding experiment, is of the order of tens of kJ/mol".

These changes are highlighted in pink.

Reviewer #2 (Remarks to the Author):

I have refereed this paper twice for Nature and so am familiar with it and I assume the editors have my previous reports. I feel the work is interesting, novel and thoroughly done. It will attract a lot of interest. I am unsure about the mechanisms but the authors have addressed all the experimental queries I had that could contradict their claims of sensitivity in complex samples. THE only changes they did not make from this version compared to the last was to do with language, being more tentative regarding the mechanism and using slightly simpler, more understandable expressions. I also note the addition of the SiMoT abbreviation which I think is silly but so be it.

I am happy for this paper to be accepted.

Reply to Reviewer # 2:

We share with Reviewer #2 the happiness for this manuscript being, in principle, finally on its way to be accepted.

As to the SiMoT abbreviation sounding silly, we actually decided to choose it because of its meaning. We found that in Tagalog (the basis of the Filipino national language) it is a verb that means “to consume entirely, to devour, to pick up everything”. For instance, it is used to say “picked up to the last grain or piece from floor, soil, dish or container” (<https://www.tagalog-dictionary.com/search?word=simot>). Hence we believe that the chosen acronym may suggest that our technology can “SiMoT” till the very last molecule!

Reviewer #3 (Remarks to the Author):

The remaining minor points from my previous reviews were addressed. I am in favor of publication of this version.

Reply to Reviewer # 3:

We thank the reviewer for being in favour of the publication of our manuscript.